# Association of serum leptin and adiponectin concentrations with echocardiographic parameters and pathophysiological states in patients with cardiovascular disease receiving cardiovascular surgery

**Tatsuya Sawaguchi[1], Toshiaki Nakajima[1¤]\*, Akiko Haruyama[1], Takaaki Hasegawa[1], Ikuko Shibasaki[2], Takafumi Nakajima[1], Hiroyuki Kaneda[1], Takuo Arikawa[1], Syotaro Obi[1], Masashi Sakuma[1], Hironaga Ogawa[2], Yuusuke Takei[2], Shigeru Toyoda[1], Fumitaka Nakamura[3], Shichiro Abe[1], Hirotsugu Fukuda[2], Teruo Inoue[1]**

**1** Department of Cardiovascular Medicine, Dokkyo Medical University, Tochigi, Japan, **2** Department of Cardiovascular Surgery, Dokkyo Medical University, Tochigi, Japan, **3** Third Department of Internal Medicine, Teikyo University, Chiba Medical Center, Japan

¤ Current address: Heart Center, Dokkyo Medical University, Kitakobayashi, Mibu, Tochigi, Japan
\* nakat@dokkyomed.ac.jp

## Abstract

Leptin and adiponectin are important regulators of energy metabolism and body composition. Leptin exerts cardiodepressive effects, whereas adiponectin has cardioprotective effects, but several conflicting findings have been reported. The aim of the present study was to assess the relationship between serum leptin and adiponectin levels and echocardiographic parameters and pathophysiological states in patients with cardiovascular disease (CVD) receiving cardiovascular surgery. A total of 128 patients (79 males, average age 69.6 years) that had surgery for CVD including coronary artery bypass graft (CABG) and valve replacement were recruited in this study. Preoperative serum adiponectin and leptin concentrations were measured by enzyme-linked immunosorbent assay and compared with preoperative echocardiographic findings. Body fat volume and skeletal muscle volume index (SMI) were estimated using bioelectrical impedance analysis. We also measured grip strength and gait speed. Sarcopenia was diagnosed based on the recommendations of the Asian Working Group on Sarcopenia. Positive correlations were found between adiponectin and brain natriuretic peptide (BNP), age, left atrial diameter (LAD), E/e' (early-diastolic left ventricular inflow velocity / early-diastolic mitral annular velocity), and left atrial volume index (LAVI). Negative correlations were observed between adiponectin and body mass index (BMI), estimated glomerular filtration rate (eGFR), triglyceride, hemoglobin, and albumin. Serum leptin was positively correlated with BMI, total cholesterol, triglyceride, albumin, body fat volume, and LV ejection fraction (LVEF), whereas it was negatively correlated with BNP and echocardiographic parameters (LAD, LV mass index (LVMI), and LAVI). Multiple regression analysis showed associations between log (leptin) and log (adiponectin) and echocardiographic parameters after adjusting for age, sex, and BMI. Serum adiponectin

**Data Availability Statement:** All relevant data are within the manuscript and its Supporting Information files.

**Funding:** This study was supported in part by JSPS KAKENHI Grant Number 16H03203, and 19H03981 (to T.N.). The funding sources for this study had no role in study design, data collection, analysis, or interpretation.

**Competing interests:** The authors have declared that no competing interests exist.

**Abbreviations:** CVD, cardiovascular disease; CABG, coronary artery bypass graft; AVR, aortic valve replacement; MVR, mitral valve replacement; MVP, mitral valve plasty; TVP, tricuspid valve plasty; TVR, tricuspid valve replacement; TAR, total arch replacement; LA, left atrium; LV, left ventricle; BMI, body mass index; NYHA, New York Heart Association; TNFα, tumor necrosis factor α; SMI, skeletal muscle mass index; BNP, brain natriuretic peptide; eGFR, estimated glomerular filtration rate; hsCRP, high sensitive C-reactive protein; LAD, left atrial diameter; LVDd, left ventricular end-diastolic diameter; LVDs, left ventricular end-systolic diameter; IVSth, intraventricular septum thickness; PWth, posterior wall thickness; LVEF, LV ejection fraction; LVM, LV mass; LVMI, LV mass index; E, early-diastolic left ventricular inflow velocity; e', early-diastolic mitral annular velocity; LAVI, left atrial mass index; ROC curve, receiver operating characteristic curve; AUC, area under the curve.

was negatively correlated with leptin, but positively correlated with tumor necrosis factor α (TNFα), an inflammatory cytokine. In males, serum leptin level had a positive correlation with skeletal muscle volume and SMI. However, adiponectin had a negative correlation with anterior mid-thigh muscle thickness, skeletal muscle volume and SMI. And, it was an independent predictive factor in males for sarcopenia even after adjusted by age. These results suggest that leptin and adiponectin may play a role in cardiac remodeling in CVD patients receiving cardiovascular surgery. And, adiponectin appears to be a marker of impaired metabolic signaling that is linked to heart failure progression including inflammation, poor nutrition, and muscle wasting in CVD patients receiving cardiovascular surgery.

## Introduction

Adiponectin and leptin are well known to play important roles in regulating metabolic homeostasis and are linked to several pathophysiological conditions and diseases including cardiovascular disease (CVD). Adiponectin with anti-inflammatory, insulin-sensitizing, and anti-atherogenic properties, is a cardioprotective adipokine synthesized and secreted in large quantities from adipose tissue [1,2]. High levels of circulating adiponectin are known to have a favorable effect on metabolic processes and protect against derangements that lead to obesity, metabolic syndrome, atherosclerosis, and subsequently CVD [1–3]. In contrast, obesity is associated with high serum leptin levels (hyperleptinemia) and low serum adiponectin levels (hypoadiponectinemia) [4,5], and leptin has been generally thought to be a cause of many types of CVD associated with obesity [6].

Adiponectin and leptin also play an important role in regulating cardiac function and are linked to several cardiac pathophysiological conditions and diseases, particularly cardiac hypertrophy and heart failure. Adiponectin induces anti-apoptotic effects, reduces fibrosis and oxidative stress in the myocardium [7–9], and low adiponectin levels correlate with left ventricular hypertrophy and diastolic dysfunction [10–13]. In contrast to the cardioprotective effects of adiponectin, high adiponectin levels have also been reported to be associated with increased risk of recurrent cardiovascular events [14] and mortality in patients with acute myocardial infarction [15]. Furthermore, a paradoxical increase in circulating adiponectin levels has also been reported in patients with systolic and diastolic heart failure, and higher plasma concentrations of the hormone are associated with worse prognosis in heart failure [16–18], and cardiovascular surgery patients [19]. As a possible mechanism, cardiac cachexia and muscle wasting (sarcopenia) lead to increased levels of adiponectin in heart failure [20,21].

Obesity is known to be associated with larger left atrial (LA) size, left ventricular (LV) mass and wall thickness, and diastolic dysfunction [22,23,24], and obesity increases the risk of heart failure [25]. A few experimental studies have also reported the adverse cardiac effects of leptin on isolated cardiomyocytes including the promotion of muscle hypertrophy [26]. On the other hand, recent studies have shown that leptin may have cardioprotective effects [27–29]. They reported that higher leptin was associated with lower LV mass, wall thickness, and LA size in individuals older than 70 years of age and subjects without CVD. Experimental studies have also reported worse cardiac function and prognosis after experimentally-induced myocardial infarction in leptin-deficient mice [30]. In addition, Barouch et al. [31] showed that disruption of leptin signaling contributed to cardiac hypertrophy independently of body weight. Thus, the pathophysiological significance of leptin and adiponectin has not been clarified in patients with CVD. We have recently described the associations of the adipokine (adiponectin and

leptin) concentrations with epicardial fat volume in cardiovascular surgery patients [32]. How-ever, few studies have evaluated the associations of serum leptin and adiponectin concentra-tions with echocardiographic parameters and pathophysiological states in CVD patients receiving cardiovascular surgery.

The aim of the present study was to clarify the pathophysiological roles of circulating serum leptin and adiponectin in patients with cardiovascular disease (CVD) receiving cardiovascular surgery. Here, we investigated the relationships between serum leptin and adiponectin levels and echocardiographic findings and pathophysiological states as well as clinical laboratory data.

## Materials and methods

### Participants

A total of 128 patients aged 23–89 years (69.6 ± 12.6 years) undergoing cardiovascular surgery at Dokkyo Medical Hospital were included in this study. Table 1 shows the characteristics of the patients. Seventy-nine patients were males (62%) and 49 patients were females (38%). The body mass index (BMI) was 23.6 ± 4.1 kg/m$^2$. The average preoperative New York Heart Asso-ciation (NYHA) classification was 2.2 ± 1.0. Most of the patients had conventional risk factors such as hypertension (HT), diabetes (DM), hyperlipidemia (HL), current smoking, and hemo-dialysis (HD). Table 1 also shows the number of patients classified by surgical procedures for cardiovascular disease. The study was approved by the Ethics Committee of the Dokkyo Medi-cal University (No. 27077), and written informed consents were obtained from all participants. All patients had medical treatment including β-blocking agents, calcium-channel blockers, angiotensin receptor blockers (ARB)-/-angiotensin converting enzyme inhibitors (ACEI), diuretics, statins, and anti-diabetic drugs (Table 1).

The fasting total cholesterol (T-Chol), hemoglobin A1 (HbA1c), albumin (Alb), brain natri-uretic peptide (BNP), low-density lipoprotein (LDL)-cholesterol (LDL-C), high density lipo-protein (HDL)-cholesterol (HDL-C), non-HDL-C, triglycerides (TG), and estimated glomerular filtration rate (eGFR) were measured before the operation. The biochemical data were analyzed using routine chemical methods in the Dokkyo Medical University Hospital clinical laboratory. Levels of the inflammatory marker, high-sensitivity C-reactive protein (hsCRP), were measured by a latex-enhanced nephelometric immunoassay (N Latex CRP II and N Latex SAA, Dade Behring Ltd., Tokyo, Japan).

To measure fasting serum adiponectin, leptin, and tumor necrosis factor α (TNFα) levels, peripheral venous blood was drawn into pyrogen-free tubes with and without EDTA on the morning of cardiovascular surgery. For plasma, the EDTA tubes were placed on melting ice, subsequently centrifuged with 20 min at 1500g for 10 min at 4˚C. Plasma and serum were stored in aliquots at -80˚C for all enzyme-linked immunosorbent assays (ELISA).

### Transthoracic echocardiography

Each patient received preoperative transthoracic echocardiography. Two-dimensional (2D) images were recorded with an iE33 and EPICQ7 cardiovascular ultrasound system (PHILIP, Amsterdam, Netherlands) with a 1.7–3.4 MHz Doppler transducer. 2D echocardiography was performed according to the recommendations of the American Society of Echocardiography. Left atrial diameter (LAD), left ventricular end-diastolic diameter (LVDd), left ventricular end-systolic diameter (LVDs), interventricular septal thickness (IVSth), and LV posterior wall thickness (PWth) were measured using the parasternal long-axis view. Left ventricular mass (LV mass, LVM) was estimated by LVDd and wall thickness (IVSth and PWth) and then

**Table 1. Patient characteristics.**

| | |
|---|---|
| **Total patients (number)** | **128** |
| Male / Female | 79 / 49 |
| Age, years | 69.6 ± 12.6 |
| BMI, kg/m$^2$ | 23.6 ± 4.1 |
| Atrial fibrillation, number | 42 |
| NYHA | 2.2 ± 1.0 |
| Risk factors, number | 128 |
| Hypertension | 86 |
| Diabetes | 28 |
| Dyslipidemia | 61 |
| Smoking | 19 |
| Hemodialysis | 9 |
| Cardiovascular surgery, number | 128 |
| CABG | 29 |
| AVR | 21 |
| MVR (MVP) with or without TVR (TAP) | 24 |
| CABG combined with valve replacement / repair (AVR, MVP,TAP) | 11 |
| AVR combined with other valve (MVP, TAP) or aortic diseases (TAR) | 25 |
| Aortic disease (TAR, TEVAR, et cetra) | 11 |
| Others | 7 |
| Drugs, number | 128 |
| β-blockers | 61 |
| Ca-blockers | 42 |
| ACE-I/ARB | 72 |
| Diuretics | 62 |
| Statin | 60 |
| Anti-diabetic drugs | 25 |
| Echocardiographic findings | |
| LAD, mm | 43.5 ± 9.1 (116) |
| LVDd, mm | 51.9 ± 10.5 (117) |
| LVDs, mm | 36.0 ± 9.8 (117) |
| LVEF, % | 57.7 ± 12.6 (116) |
| LVMI, g/m$^2$ | 113.0 ± 39.0 (114) |
| E/e' | 20.3 ± 11.2 (110) |
| LAVI, ml/m$^2$ | 41.4 ± 26.4 (107) |

The data are shown as the mean ± SD or the number of patients with a certain characteristic; (number): number of patients examined.

BMI, body mass index; NYHA, New York Heart Association; CABG, coronary artery bypass grafting; AVR, aortic valve replacement; MVR, mitral valve replacement; MVP, mitral valve plasty; TVP, tricuspid valve plasty; TVR, tricuspid valve replacement; TAR, total arch replacement; TEVAR, thoracic endovascular aortic repair; ACE-I, angiotensin converting enzyme inhibitor; ARB, angiotensin II receptor blocker; Antidiabetic drugs (i.e,. α-glucosidase inhibitor, sulfonylurea, biguanide, dipeptidyl peptidase-4 inhibitor, sodium glucose cotransporter 2 inhibitor); LAD, left atrial diameter; LVDd, left ventricular end-diastolic diameter; LVDs, left ventricular end-systolic diameter; LVEF, ejection fraction; LVMI, left ventricular mass index; E/e', the ratio of early-diastolic left ventricular inflow velocity (E) to early-diastolic mitral annular velocity (e'): LAVI, left atrial volume index

indexed to body surface area (LVMI).

$$LVmass = 0.8[1.04\{(LVDd + IVSth + PWth)^3 - LVDd^3\}] + 0.6$$

Left ventricular end-diastolic volume (LVEDV) and end-systolic volume (LVESV) were measured from the apical view with the biplane method. Left ventricular ejection fraction (LVEF) was calculated using the Simpson method.

$$LVEF = 100(LVEDV - LVESV)/LVEDV$$

Doppler echocardiography was performed for detecting early and late diastolic transmitral flow velocity and E/A was calculated. E/e' was determined by the ratio of early-diastolic left ventricular inflow velocity (E) to early-diastolic mitral annular velocity (e'). LA volume (LAV) was measured from the parasternal long-axis view (LA1) and the apical four-chamber view (LA2 and LA3) with the prolate ellipse method and then indexed to body surface area (LAVI).

$$LAV = \pi\{(LA1)(LA2)(LA3)\}/6$$

### Enzyme-linked immunosorbent assay (ELISA)

Serum adiponectin level was measured by the Human Total Adiponectin/Acrp30 Quantikine ELISA Kit (DRP300, R&D Systems, Minneapolis, MN, USA), as described previously [32,33]. The detection threshold was 0.24 ng/ml. Samples, reagents, and buffers were prepared according to the manufacturer's instructions. Serum leptin levels were also measured by the human Quantikine ELISA Kit (DLP00 for leptin, R&D Systems, Minneapolis, MN, USA). The detection thresholds of leptin were 7.8 pg/ml. The serum concentration of tumor necrosis factorα (TNFα) was measured by the Human Quantikine HS ELISA Kit (HSTA00E, R&D Systems, Minneapolis, MN, USA), and the detection threshold was 0.022 pg/ml.

### Measurements of gait speed, grip strength, and voluntary isometric contraction

The gait speed was measured as the time needed to walk 4 m at an ordinary pace. Maximal voluntary isometric contraction (MVIC) of the knee extensors was measured using a digital hand-held dynamometer (μTas MT-1, ANIMA Co., Ltd., Tokyo, Japan), as described previously [34]. Each subject performed 2 trials, and the highest score was adopted for analysis.

### Measurements of bioelectrical impedance

A multi-frequency bioelectrical impedance analyzer (BIA), InBody S10 Biospace device (Biospace Co, Ltd, Korea/Model JMW140) was used according to the manufacturer's guidelines, as described previously [34]. Thirty impedance measurements were obtained using 6 different frequencies (1, 5, 50, 250, 500, and 1000 kHz) at the 5 following segments of the body (right and left arms, trunk, and right and left legs). The measurements were carried out while the subjects rested quietly in the supine position, with their elbows extended and relaxed along their trunk. Body fat volume, body fat percentage, and skeletal muscle volume were measured. Also, skeletal muscle mass index (SMI; appendicular skeletal muscle mass/height$^2$, kg/m$^2$) was measured as the sum of lean soft tissue of the two upper limbs and two lower limbs. In this study, sarcopenia was defined according to the Asian Working Group for Sarcopenia (AWGS) criteria (age $\geq$ 65 years; handgrip $<$ 26 kg for males and $<$ 18 kg for females or gait speed $\leq$ 0.8 m/sec, and SMI $<$ 7.0 kg/m$^2$ for males and $<$ 5.7 kg/m$^2$ for females) [35].

## Measurement of muscle thickness by ultrasound

The anterior mid-thigh muscle thickness was measured on the right leg using a real-time linear electronic scanner with a 10.0-MHz scanning head and Ultrasound Probe (L4–12t-RS Probe, GE Healthcare Japan) and LOGIQ e ultrasound (GE Healthcare Japan), as previously described [34]. From the ultrasonic image, the subcutaneous adipose tissue-muscle interface and the muscle-bone interface were identified. The perpendicular distance from the adipose tissue-muscle interface to the muscle-bone interface was considered to represent the anterior thigh muscle thickness (TMth). The measurement was performed twice in both the supine and standing positions, and the average value was adopted for analysis.

## Statistical analysis

Data are presented as mean value ± SD. After testing for normality (Kolmogorov-Smirnov), the comparison of means between groups was analyzed by a two-sided, unpaired Student's t-test in the case of normally distributed parameters or by the Mann-Whitney-U-Test in the case of non-normally distributed parameters. Associations among parameters were evaluated using Pearson or Spearman correlation coefficients. A receiver operating characteristic (ROC) curve was plotted to identify the optimal cut-off level of the serum concentration of adiponectin for detecting sarcopenia. Multiple linear regression analysis with serum adiponectin, leptin concentration or echocardiographic parameters as the dependent variable was performed to identify independent predictors (clinical laboratory data, echocardiographic parameters, or physical data) of serum leptin, adiponectin levels, or echocardiographic parameters. Age, sex, and BMI were employed as covariates. When the independent or dependent data were not normally distributed, the data were logarithmically transformed to achieve a normal distribution. Logistic regression analysis was used to identify serum factors (leptin and adiponectin) independently associated with sarcopenia. All analyses were performed using SPSS version 24 (IBM Corp., New York, USA) for Windows. A $p$ value of $\leq 0.05$ was regarded as significant.

## Results

### Characteristics of the patients

The clinical characteristics and sex differences of the study patients are shown in Tables 1 and 2. The mean age of females was higher than that of males (67.3 ± 13.1 years vs. 73.0 ± 10.9 years, $p < 0.05$). The gait speed, grip strength, knee extension strength, and SMI in females were significantly lower than those in males, but the body fat percentage was higher in females. The levels of HDL-C and T-Chol in females was higher than that in males (HDL-C, 49.9 ± 15.3 mg/dl vs. 59.1 ± 17.7 mg/dl, $p < 0.01$; T-Chol, 161 ± 38.4 mg/dl vs. 178 ± 35.4 mg/dl, $p < 0.05$). The serum adiponectin level in females was significantly higher than that in males (11.5 ± 7.4 μg/ml vs. 8.3 ± 7.1 μg/ml, $p < 0.05$). The serum leptin level was 4855 ± 6975 pg/ml in all of the patients. It was higher in females than in males (7865 ± 9139 pg/ml vs. 2929 ± 4286 pg/ml, $p < 0.01$). The serum TNPα level was higher in males than in females (1.31 ± 0.65 pg/ml vs. 1.12 ± 0.71 pg/ml, $p < 0.05$).

### Correlation between serum leptin and adiponectin levels and the clinical data

The correlations between serum leptin and adiponectin concentrations and the clinical data are shown in total patients (males and females) as shown in Table 3. The serum leptin level was not correlated with age, whereas the serum adiponectin concentration was positively correlated with age ($r = 0.385$, $p = 0.000$). The serum leptin level was strongly positively correlated

**Table 2. Sex differences in various parameters.**

| | Total (n = 128) | Male (n = 79) | Female (n = 49) |
|---|---|---|---|
| Age, years | 69.5 (12.6) | 67.3 (13.1) | **73.0 (10.9)**[*] |
| BMI, kg/m$^2$ | 23.7 (4.0) | 23.9 (4.0) | 23.4 (3.9) |
| NYHA | 2.2 (1.0) | 2.2 (1.1) | 2.1 (0.9) |
| Gait speed, m/s | 0.92 (0.31) [92] | 0.98 (0.31) [55] | **0.84 (0.29)**[*] [37] |
| Grip strength, kgf | 23.0 (8.8) [94] | 27.1 (8.2) [56] | **16.8 (5.4)**[***] [38] |
| Knee extension strength, kgf | 21.4 (10.1) [88] | 24.6 (10.5) [52] | **16.7 (7.3)**[***] [36] |
| Body fat percentage, % | 31.2 (9.1) [106] | 27.2 (7.4) [66] | **37.7 (7.6)**[***][40] |
| Skeletal muscle mass index (SMI), kg/m$^2$ | 6.4 (1.3) [104] | 7.1 (1.1) [65] | **5.4 (0.9)**[***] [39] |
| Anterior mid-thigh muscle thickness, cm | 2.28 (0.73) [89] | 2.37 (0.78) [53] | 2.16 (0.64) [36] |
| HbA1c, % | 6.1 (0.8) | 6.2 (0.9) | 5.9 (0.7) |
| BNP, pg/ml | 383 (583) | 377 (560) | 389 (622) |
| eGFR, ml/min/1.73 m$^2$ | 60.0 (33.0) | 57.0 (29.6) | 64.9 (37.7) |
| Hb, g/dl | 12.3 (2.0) | 12.4 (2.2) | 11.9 (1.6) |
| LDL-C, mg/dl | 92.2 (27.6) | 89.5 (27.9) | 97.1 (26.8) |
| HDL-C, mg/dl | 53.5 (16.8) | 49.9 (15.3) | **59.1 (17.7)**[**] |
| Non HDL-C, mg/dl | 114 (31.9) | 111 (32.9) | 119 (29.9) |
| T-Chol, mg/dl | 168 (38.0) | 161 (38.4) | **178 (35.4)**[*] |
| TG, mg/dl | 107 (63) | 106 (61) | 109 (66) |
| Albumin (Alb), g/dl | 3.8 (0.62) | 3.8 (0.62) | 3.9 (0.61) |
| hsCRP, mg/dl | 0.85 (1.7) | 0.97 (2.0) | 0.64 (1.2) |
| Adiponectin, μg/ml | 9.5 (7.3) | 8.3 (7.1) | **11.5 (7.4)**[*] |
| Leptin pg/ml | 4855 (6975) | 2929 (4286) | **7865 (9139)**[**] |
| TNFα, pg/ml | 1.23 (0.68) | 1.31 (0.65) | **1.12 (0.71)**[*] |

[*]$p < 0.05$

[**]$p < 0.01$ for male vs. female. [number]; number of patients that could be evaluated. The data are shown as the mean (SD). BNP, brain natriuretic peptide; eGFR, estimated glomerular filtration rate; hsCRP, high-sensitivity C-reactive protein; TNFα, tumor necrosis factor α; TG, triglycerides; T-Chol, total cholesterol; HDL-C, high density lipoprotein cholesterol; LDL-C, low density lipoprotein cholesterol; Hb, hemoglobin

with BMI ($r = 0.525$, $p = 0.000$), whereas the adiponectin level was negatively correlated with BMI ($r = -0.359$, $p = 0.000$). The concentration of leptin was negatively correlated with BNP ($r = -0.300$, $p = 0.002$,), but the adiponectin level was strongly positively correlated with BNP ($r = 0.628$, $p = 0.000$). The leptin level was positively correlated with T-Chol ($r = 0.374$, $p = 0.000$), TG ($r = 0.325$, $p = 0.000$), LDL-C ($r = 0.354$, $p = 0.000$), and non-HDL-C ($r = 0.408$, $p = 0.000$). The concentration of adiponectin was negatively correlated with eGFR ($r = -0.300$, $p = 0.001$), and TG ($r = -0.468$, $p = 0.000$), non-HDL-C ($r = -0.242$, $p = 0.007$), and HbA1C ($r = -0.218$, $p = 0.017$), whereas it was positively correlated with HDL-C ($r = 0.221$, $p = 0.015$). Leptin was positively correlated with albumin (Alb) ($r = 0.361$, $p = 0.000$), whereas adiponectin was negatively correlated with hemoglobin (Hb) ($r = -0.276$, $p = 0.002$), and Alb ($r = -0.218$. $p = 0.014$). Leptin did not correlated with serum TNFα level, while adiponectin had a positive correlation with the TNFα level ($r = 0.301$, $p = 0.001$). The serum level of adiponectin was negatively correlated with leptin ($r = -0.207$, $p = 0.020$), and positively correlated with TNFα, an inflammatory cytokine ($r = 0.301$, $p = 0.001$).

The linear regression analysis with serum adiponectin levels as the dependent variable and clinical data as independent variable were investigated in all of the patients. As shown in Table 4, multiple regression analysis showed that log (BNP) (β = 0.294, $p = 0.002$), eGFR (β =

**Table 3. Correlation matrix between clinical data and serum adiponectin and leptin levels.**

| | Leptin<br>r—value (p—value) | Adiponectin<br>r—value (p—value) |
|---|---|---|
| Age | -0.042 (0.637) | **0.385 (0.000)**\*\*\* |
| BMI | **0.525 (0.000)**\*\*\* | **-0.359 (0.000)**\*\*\* |
| BNP | **-0.300 (0.002)**\*\* | **0.628 (0.000)**\*\*\* |
| eGFR | -0.148 (0.096) | **-0.300 (0.001)**\*\* |
| T-Chol | **0.374 (0.000)**\*\*\* | -0.060 (0.512) |
| TG | **0.325 (0.000)**\*\*\* | **-0.468 (0.000)**\*\*\* |
| HDL-C | 0.122 (0.177) | **0.221 (0.015)**\* |
| LDL-C | **0.354 (0.000)**\*\*\* | -0.060 (0.512) |
| Non-HDL-C | **0.408 (0.000)**\*\*\* | **-0.242 (0.007)**\*\* |
| HbA1C | 0.086 (0.345) | **-0.218 (0.017)**\* |
| Hb | 0.091 (0.306) | **-0.276 (0.002)**\*\* |
| Alb | **0.361 (0.000)**\*\*\* | **-0.218 (0.014)**\* |
| hsCRP | -0.112 (0.208) | 0.090 (0.317) |
| TNFα | -0.152 (0.090) | **0.301 (0.001)**\*\* |
| Leptin | - | **-0.207 (0.020)**\* |
| Echocardiographic parameters | | |
| LAD | **-0.310 (0.001)**\*\* | **0.352 (0.000)**\*\*\* |
| LVDd | **-0.247 (0.007)**\*\* | -0.078 (0.410) |
| LVDs | **-0.298 (0.001)**\*\* | 0.016 (0.864) |
| IVSth | 0.140 (0.132) | -0.124 (0.185) |
| PWth | 0.084 (0.371) | -0.078 (0.406) |
| LVEF | **0.203 (0.029)**\* | -0.175 (0.063) |
| LVM | **-0.206 (0.028)**\* | -0.091 (0.388) |
| LVMI | **-0.287 (0.002)**\*\* | 0.025 (0.791) |
| E/e' | -0.120 (0.210) | **0.353 (0.000)**\*\*\* |
| LAVI | **-0460 (0.000)**\*\*\* | **0.538 (0.000)**\*\*\* |

\* $p < 0.05$

\*\* $p < 0.01$

\*\*\* $p < 0.001$ LAD, left atrial diameter; LVDd, left ventricular end-diastolic diameter; LVDs, left ventricular end-systolic diameter; IVSth, intraventricular septum thickness; PWth, posterior wall thickness; LVEF, LV ejection fraction; LVM, LV mass; LVMI, LV mass index; E/e', the ratio of early-diastolic left ventricular inflow velocity (E) to early-diastolic mitral annular velocity (e'): LAVI, left atrial mass index; other abbreviations as in Table 2.

-0.237, $p = 0.005$), log (TG) ($β = -0.193$, $p = 0.044$), and log (HDL-C) ($β = 0.175$, $p = 0.038$) were the independent variables to predict log (serum adiponectin concentration) after adjusting for age, sex, and BMI. On the other hand, log (Alb) ($β = 0.292$, $p = 0.003$) was the independent variable to predict log (serum leptin concentration) after adjusting for age, sex, and BMI (Table 5).

**Table 4. Multiple linear regression analysis of serum adiponectin concentrations and the clinical data.** Dependent variable: adiponectin (log).

| Independent variable | BNP (log) | eGFR | TG (log) | HDL-C (log) | Non-HDL-C | HbA1c | Hb | Albumin (log) |
|---|---|---|---|---|---|---|---|---|
| β-value (p) | **0.294 (0.002)**\*\* | **-0.237 (0.005)**\* | **-0.193 (0.044)**\* | **0.175 (0.038)**\* | -0.120 (0.220) | -0.146 (0.054) | -0.030 (0.763) | -0.052 (0.616) |

\* $p < 0.05$

\*\*$p < 0.01$ Model, adjusted by age, sex, and BMI

**Table 5. Multiple linear regression analysis of serum leptin concentration and the clinical data.** Dependent variable: leptin (log).

| Independent variable | BNP (log) | TG (log) | Non-HDL-C | HbA1C | Hb | Albumin (log) |
|---|---|---|---|---|---|---|
| β-value ($p$) | -0.010 (0.900) | 0.093 (0.290) | 0.088 (0.313) | 0.105 (0.144) | -0.113 (0.237) | **0.292 (0.003)**[**] |

[**]$p < 0.01$ Model, adjusted by age, sex, and BMI

## Correlations between serum adiponectin and leptin levels and the echocardiographic findings

Table 3 also show the relationships between serum adiponectin and leptin levels and the echocardiographic findings in all of the patients. The baseline echocardiographic data are shown in Table 1. The serum leptin level was negatively correlated with LAD ($r$ = -0.310, $p$ = 0.001, S1Aa Fig), LVDd, LVDs, LVM, and LVMI. It was also negatively correlated with LAVI ($r$ = -0.460, $p$ = 0.000, S1Ba Fig), but not with E/e' ($r$ = -0.120, $p$ = 0.210, S1Ca Fig). It was positively correlated with LVEF ($r$ = 0.203, $p$ = 0.029). On the other hand, the serum adiponectin level was positively correlated with LAD ($r$ = 0.352, $p$ = 0.000, S1Ab Fig), LAVI ($r$ = 0.538, $p$ = 0.000, S1Bb Fig), and E/e' ($r$ = 0.353, $p$ = 0.000, S1Cb Fig).

The linear regression analysis with serum leptin levels as the dependent variable and echocardiographic data as independent variable were investigated (Table 6A). Multiple regression analysis showed that LAD (β = -0.262, $p$ = 0.002), and log (LVMI) (β = -0.166, $p$ = 0.036) were the independent variables to predict serum leptin concentration after adjusting for age, sex, and BMI. On the other hand, the linear regression analysis with echocardiographic parameters (LAVI, E/e' and LVMI) as dependent variable and serum leptin and adiponectin levels as independent variable were investigated in all of the patients. As shown in Table 6B, both leptin and adiponectin level were the independent variable (β = -0.434, $p$ = 0.000 for leptin, β = 0.398, $p$ = 0.000 for adiponectin) to predict LAVI after adjusting for age, sex, and BMI. In addition, adiponectin level was the independent variable to predict E/e' (β = 0.245, $p$ = 0.022) (Table 6C), while leptin level was the independent variable to predict LVMI (β = 0.370, $p$ = 0.003) after adjusting for age, sex, and BMI (Table 6D).

## Correlations between serum adiponectin and leptin concentrations, physical function and the BIA findings

Table 7 shows the relationships between serum adipokine (leptin and adiponectin) levels, physical function, and the BIA findings in both males and females. The serum leptin level was strongly positively correlated with body fat volume and body fat percentage in both males and females. In males, it had a positive correlation with skeletal muscle volume and SMI. On the other hand, adiponectin had a negative correlation with anterior mid-thigh muscle thickness, skeletal muscle volume and SMI in men, and it tended to correlate with body fat volume. In females, it had a weak negative correlation with SMI, and it tended negatively to correlate with skeletal muscle volume, and body fat volume.

Logistic regression analysis is used to obtain odds ratio in the presence of the explanatory variable (serum leptin and adiponectin levels) for sarcopenia in both males and females. Serum adiponectin level was an independent predictive factor in males ($p$ = 0.037; odds ratio, 1.119; 95% confidence interval [CI] 1.007–1.124) for sarcopenia even after adjusted by age. However, serum leptin level was not an independent predictor factor ($p$ = 0.440). In females, neither adiponectin nor leptin level was an independent predictive factor ($p$ = 0.955 for adiponectin: $p$ = 0.405 for leptin) for sarcopenia, adjusted by age. An ROC curve was plotted to

**Table 6. Multiple linear regression analysis of the serum leptin and adiponectin concentration and the echocardiographic findings.**

| A) Dependent variable: leptin (log) | Model 1 | Model 2 | Model 3 | Model 4 |
|---|---|---|---|---|
| Independent variable | β-value (*p*) | β-value (*p*) | β-value (*p*) | β-value (*p*) |
| LAD | **-0.310 (0.002)**\*\* | **-0.327 (0.001)**\*\* | **-0.306 (0.002)**\*\* | **-0.262 (0.002)**\*\* |
| EF (log) | 0.123 (0.172) | 0.120 (0.184) | 0.064 (0.476) | -0.021 (0.792) |
| LVMI (log) | **-0.215 (0.024)**\* | **-0.213 (0.026)**\* | **-0.200 (0.030)**\* | **-0.166 (0.036)**\* |
| E/e' (log) | -0.028 (0.753) | -0.014 (0.878) | -0.045 (0.612) | 0.010 (0.893) |
| B) Dependent variable: LAVI (log) | Model 1 | Model 2 | Model 3 | Model 4 |
| Independent variable | β-value (*p*) | β-value (*p*) | β-value (*p*) | β-value (*p*) |
| Leptin (log) | **-0.382 (0.000)**\*\*\* | **-0.379 (0.000)**\*\*\* | **-0.424 (0.000)**\*\*\* | **-0.434 (0.000)**\*\*\* |
| Adiponectin (log) | **0.373 (0.000)**\*\*\* | **0.426 (0.000)**\*\*\* | **0.397 (0.000)**\*\*\* | **0.398 (0.000)**\*\*\* |
| C) Dependent variable: E/e' (log) | Model 1 | Model 2 | Model 3 | Model 4 |
| Independent variable | β-value (*p*) | β-value (*p*) | β-value (*p*) | β-value (*p*) |
| Leptin (log) | -0.014 (0.878) | -0.016 (0.863) | -0.049 (0.637) | 0.013 (0.918) |
| Adiponectin (log) | **0.306 (0.002)**\*\* | **0.275 (0.008)**\*\* | **0.255 (0.016)**\* | **0.245 (0.022)**\* |
| D) Dependent variable: LVMI (log) | Model 1 | Model 2 | Model 3 | Model 4 |
| Independent variable | β-value (p) | β-value (p) | β-value (p) | β-value (p) |
| Leptin (log) | -0.331 (0.001)\*\* | -0.327 (0.001)\*\* | -0.306 (0.003)\*\* | -0.370 (0.003)\*\* |
| Adiponectin (log) | -0.005 (0.956) | 0.020 (0.845) | 0.032 (0.758) | 0.041 (0.692) |

\* *p* < 0.05

\*\* *p* < 0.01

\*\*\* *p* < 0.001 Model 1, unadjusted; Model 2, adjusted by age, Model 3, adjusted by age and sex, Model 4, adjusted by age, sex, and BMI

identify the optimal cut-off level of the serum adiponectin concentration to detect sarcopenia in males, as shown in Fig 1. To generate the ROC curve, different adiponectin cut-off values were used to predict sarcopenia with true positives on the vertical axis (sensitivity) and false-positives (1-specificity) on the horizontal axis. The area under the curve (AUC) was 74.7%. Sensitivity and specificity were 84.2%, and 64%, respectively. The optimal cut-off value was 6.2 μg/ml.

**Table 7. Relationships between serum adipokine (leptin and adiponectin) levels, physical function, and the BIA findings in both males and females.**

| Physical & BIA findings | Leptin Males / Females<br>*r*—value (*p*—value) | Adiponectin Males / Females<br>*r*—value (*p*—value) |
|---|---|---|
| Grip strength | -0.160 (0.243) / -0.128 (0.463) | -0.186 (0.181) / -0.110 (0.529) |
| Knee extension | 0.015 (0.917) / 0.251 (0.167) | -0.201 (0.165) / -0.069 (0.708) |
| Gait speed | -0.028 (0.842) / -0.204 (0.254) | -0.163 (0.247) / -0.072 (0.690) |
| Anterior mid-thigh muscle thickness | 0.149 (0.293) / 0.347 (0.052) | **-0.364 (0.009)**xs\*\* / -0.294 (0.102) |
| Skeletal muscle volume | **0.271 (0.033)**\* / 0.025 (0.884) | **-0.355 (0.005)**\*\* / -0.292 (0.084) |
| SMI | **0.333 (0.008)**\*\* / 0.107 (0.533) | **-0.351 (0.006)**\*\* / **-0.333 (0.047)**\* |
| Body fat volume | **0.669 (0.000)**\*\*\*/ **0.630 (0.000)**\*\*\* | -0.225 (0.084) / -0.287 (0.089) |
| Body fat percentage | **0.620 (0.000)**\*\*\* / **0.643 (0.000)**\*\*\* | -0.125 (0.342) / -0.184 (0.282) |

\* *p* < 0.05

\*\* *p* < 0.01

\*\*\* *p* < 0.001 SMI, skeletal muscle mass index; BIA, bioelectric impedance analyzer; other abbreviations as in Table 2.

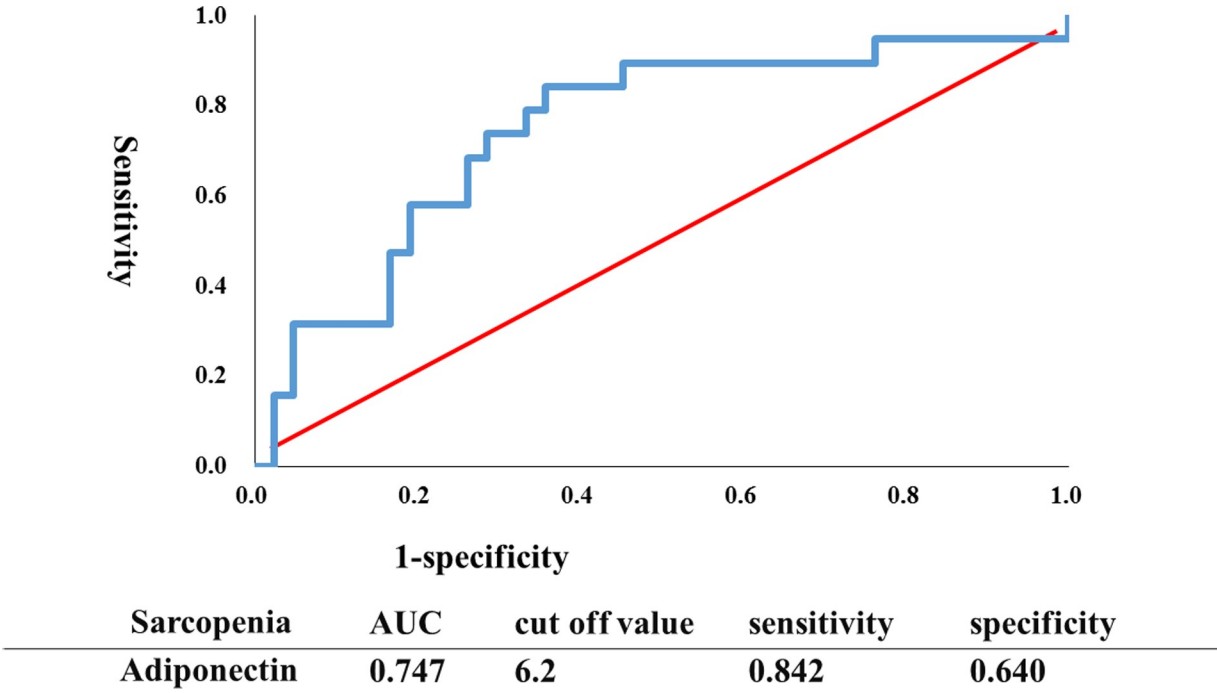

**Fig 1. ROC curve to identify the optimal cut-off level of the serum concentration of adiponectin to detect sarcopenia in males.** To generate the ROC curve shown, different adiponectin cut-off values were used to predict sarcopenia with true positives on the vertical axis (sensitivity) and false positives (1-specificity) on the horizontal axis.

Table 8 summarizes the comparative data from patients with low adiponectin levels (< 6.2 µg/ml, the cut-off value from the above ROC curve) and those with high adiponectin levels (> 6.2 µg/ml). As shown in Table 8, patients with high adiponectin levels had lower eGFR, Hb, TG, LVEF, grip strength, knee extension strength, and SMI, compared to those with low adiponectin levels. On the other hand, they had higher BNP, LAD, E/e', and LAVI. The serum concentrations of TNFα in patients with high adiponectin levels were significantly greater than those in patients with low adiponectin levels (TNFα: 1.50 ± 0.60 pg/ml vs. 1.00 ± 0.42 pg/ml, $p < 0.01$). There were no significant differences of serum leptin level in both groups.

## Discussion

The major findings of the present study are as follows. 1) In CVD patients receiving cardiovascular surgery with a mean BMI value of 23.6, the serum leptin concentration was positively correlated with LVEF, albumin, and body fat volume as well as BMI, whereas it was negatively correlated with BNP, and the echocardiographic parameters (LAD, LVMI, and LAVI). 2). The serum adiponectin level was positively correlated with BNP levels, LAD, E/e', and LAVI, but not LVEF. Negative correlations were observed between adiponectin and BMI, Hb, and albumin. 3) Multiple regression analysis showed an association between leptin or adiponectin levels and echocardiographic parameters after adjusting for age, sex, and BMI. 4) Serum adiponectin was negatively correlated with leptin, but positively correlated with TNFα. 5) In males, serum leptin level had a positive correlation with skeletal muscle volume and SMI, while adiponectin had a negative correlation with anterior mid-thigh muscle thickness, skeletal muscle volume, and SMI. And, the adiponectin level was an independent predictive factor in males for sarcopenia even after adjusted by age. These results suggest that leptin and

**Table 8. Comparison between patients with low and high adiponectin levels based on the cut-off value from the ROC curve in males.**

|  | low adiponectin group | high adiponectin group |
|---|---|---|
| Age | 62.56 (13.55) | **70.57 (12.75)**** |
| hsCRP, mg/dl | 0.62 (1.16) | 0.79 (1.65) |
| eGFR, ml/min/1.73m$^2$ | 68.38 (23.26) | **47.29 (27.03)**** |
| Hb, g/dl | 13.19 (1.78) | **11.88 (2.05)**** |
| HbA1c, % | 6.19 (0.87) | 6.16 (0.89) |
| LDL-C, mg/dl | 92.13 (25.35) | 91.21 (30.53) |
| HDL-C, mg/dl | 47.37 (11.48) | 52.85 (16.69) |
| Non-HDL-C, mg/dl | 116 (29.42) | 10.9.93 (36.96) |
| T-Chol, mg/dl | 162.93 (33.3) | 163.59 (44.24) |
| TG, mg/dl | 118.4 (55.45) | **93.25 (65.09)*** |
| Alb, g/dl | 3.92 (0.43) | 3.71 (0.60) |
| BNP, pg/ml | 94.3 (103.04) | **610.06 (688.87)**** |
| LAD, mm | 40.84 (6.42) | **49.82 (9.26)**** |
| LVEF, % | 59.05 (9.33) | **50.4 (15.38)*** |
| E/e' | 15.22 (6.78) | **19.38 (7.08)*** |
| LAVI, ml/m$^2$ | 27.50 (10.46) | **84.74 (31.37)**** |
| Grip strength, kgf | 29.79 (8.87) | **25.16 (7.31)*** |
| Knee extension, kgf | 26.64 (3.65) | **23.17 (10.25)** |
| Gait speed, m/s | 1.01 (0.28) | 0.95 (0.34) |
| SMI, kg/m$^2$ | 7.48 (0.91) | **6.82 (1.03)*** |
| Body fat volume, kg | 19.72 (8.48) | 17.46 (8.16) |
| Body fat percentage, % | 27.89 (7.87) | 26.59 (7.33) |
| Adiponectin, μg/ml | 2.86 (1.62) | **11.89 (5.53)**** |
| TNFα, pg/ml | 1.00 (0.42) | **1.50 (0.60)**** |
| Leptin | 4167 (5763) | 2376 (3138) |

The mean along with the (SD) is shown.

* $p < 0.05$

** $p < 0.01$ vs. the low adiponectin group; other abbreviations as in Table 2

adiponectin may play a role in cardiac remodeling in CVD patients receiving cardiovascular surgery. And, adiponectin appears to be a marker of impaired metabolic signaling that is linked to heart failure progression including inflammation, poor nutrition, and muscle wasting in CVD patients receiving cardiovascular surgery.

Leptin is a well-known cytokine for regulating body weight, and serum leptin levels directly correlate with body fat volume [36]. As shown previously [32], in the present study there were significant positive correlations between the serum concentration of leptin and the following metabolic risk factors: T-Chol, TG, LDL-C, non-HDL-C, BMI, body fat mass, and body fat percentage. In general, obesity is known to be associated with an increase in LA size, LVM, wall thickness, diastolic dysfunction [22,23,24], and subsequently an increased risk of heart failure [25]. However, this is somewhat opposite to the present study. The present study showed a significant inverse association of serum leptin concentration with LAD, LVMI, and LAVI in patients with CVD receiving cardiovascular surgery. Multiple regression analysis also showed that LAD and log (LVMI) were the independent variables to predict serum leptin concentration after adjusting for age, sex, and BMI. On the other hand, the leptin levels were the independent variable to predict LAVI, and LVMI after adjusting for age, sex, and BMI. And, a

significant positive correlation was observed between the serum leptin level and LVEF, but not E/e'. Furthermore, a negative correlation was observed between the serum leptin level and the plasma BNP level, a marker of heart failure severity, in contrast to the relationship between the serum adiponectin concentration and the plasma BNP level. These findings suggest that leptin favorably influences cardiac structure and function in our CVD patients, which is compatible with previous studies [27,29,37]. They reported that higher leptin was associated with lower LV mass, wall thickness, and LAD size in individuals older than 70 years of age, and general community-based patients without known cardiac disease. These contradictory findings on leptin may be partly explained by leptin resistance due to obesity, which is generally known to be associated with an increase of LAD, LVM, wall thickness, and diastolic dysfunction [22,23,24]. That is, as obesity increases, the likelihood of desensitization of the leptin receptor increases [38]. The BMI in our studies was within normal limits ($23.6 \pm 3.8$ kg/m$^2$). Thus, the present study provides the first evidence showing that leptin may be a cardioprotective adipokine that reduces cardiac remodeling in non-obese CVD patients receiving cardiovascular surgery. Thus, lower leptin levels may be associated with the loss of the protective effects of this adipokine, as reported in patients with heart failure [39]. However, the further studies using a large number of patients are required to clarify this possibility in cardiovascular surgery patients.

In this study, serum adiponectin levels showed a significant positive correlation with age, but negative correlations were observed between adiponectin and BMI, eGFR, and TG. In addition, there was a positive correlation between serum adiponectin levels and BNP and HDL-C in our study. The association of serum adiponectin level with NYHA class and BNP levels has been reported in chronic heart failure (CHF) [16,17,20,32]. This confirms the significance of the circulating adiponectin level as a prognostic marker in patients with CHF. The positive relationship between BNP and adiponectin may be explained by a previous study that showed that natriuretic peptides enhance the production of adiponectin in human adipocytes in patients with CHF [40]. Recognition that natriuretic peptides stimulate adiponectin secretion provides a mechanism linking elevated adiponectin levels to more pronounced cardiac dysfunction and a poorer prognosis [41]. Previous studies have also reported that serum adiponectin levels were correlated with cardiac geometry and function. Some studies showed that the serum adiponectin level was elevated and inversely associated with LVEF in elderly men [42], but other studies failed to confirm this [43,44]. In addition, Unno et al. [45] reported that adiponectin levels were positively associated with diastolic dysfunction in patients with hypertrophic cardiomyopathy. In the present study, no significant association was found between adiponectin concentrations and LVEF, but adiponectin concentrations were significantly positively correlated with LAD, LAVI, and E/e', a marker of diastolic dysfunction. And, the adiponectin level was the independent variable to predict LAVI, and E/e' after adjusting for age, sex, and BMI, in contrast with leptin. This is somewhat consistent with a previous study [43] that showed a positive correlation of circulatory adiponectin concentrations with E/E', and LAVI, but not LVEF in systolic heart failure patients. Thus, the present study showed that adiponectin plays an essential role in cardiac dysfunction and may serve as a link to heart failure progression in CVD patients receiving cardiovascular surgery.

Cardiac cachexia and muscle wasting (sarcopenia) have been reported to lead to increased levels of adiponectin in heart failure [17,20,21]. Elevation of circulating adiponectin levels has also been reported as a marker of weakened skeletal muscle force [46–48], decreased muscle fiber size [49], sarcopenia, and cachexia [50–52]. The present study showed that leptin had a positive correlation with skeletal muscle volume and SMI in men. On the other hand, adiponectin had a negative correlation with anterior mid-thigh muscle thickness, skeletal muscle and SMI. In females, it had a weak negative correlation with SMI. The circulating adiponectin

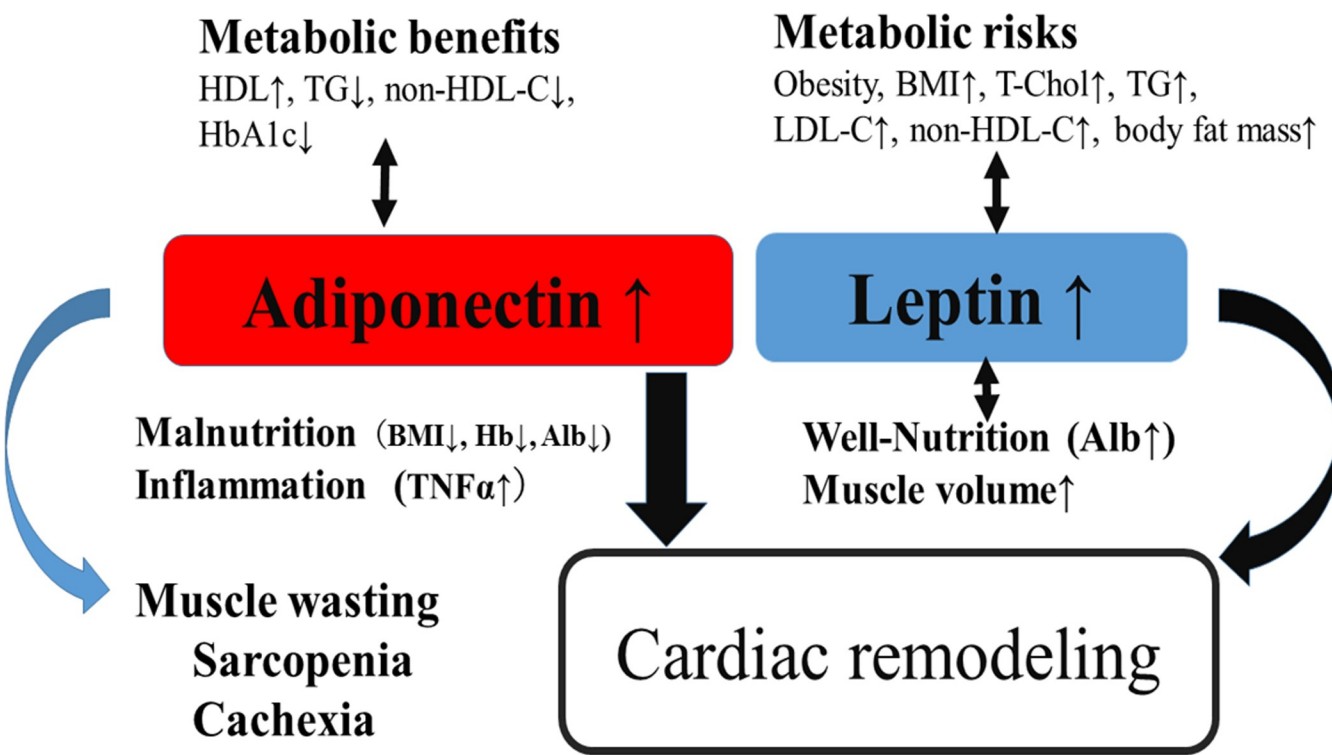

**Fig 2. A putative role of leptin and adiponectin in CVD patients receiving cardiovascular surgery.** Illustration summarizing the major findings described in the present study. Adiponectin has metabolic benefits, while leptin has metabolic risks. On the other hand, both adiponectin and leptin play a role in cardiac remodeling. Leptin appears to be associated with lower LV mass, and LA size. In contrast, adiponectin may involve diastolic dysfunction, and appears to be a marker of impaired metabolic signaling that is linked to heart failure progression including malnutrition, inflammation, and muscle wasting (cachexia, sarcopenia), especially in males, in contrast to leptin associating with well-nutrition and elevated muscle mass.

level was also inversely correlated with BMI, a well-known marker of malnutrition, and nutritional factors (Hb, albumin), suggesting that adiponectin may play a role in the pathogenesis of cachexia or sarcopenia, especially in males [17,20,21]. In addition, an ROC curve was constructed to determine the ability of adiponectin to predict sarcopenia. The AUC was 74.7% with a cut-off value of 6.2 μg/ml. This is quite similar to the results of Harada et al. [51] who did ROC curve analysis and found that the optimal cut-off value of adiponectin to detect sarcopenia in CVD patients (males and females) including cardiovascular surgery was 5.62 μg/ml. Their cut-off value was similar to the value in our study that evaluated CVD patients receiving cardiovascular surgery. In addition, the present study showed that there were positive correlations of adiponectin, but not leptin, with TNFα, an inflammatory cytokine, suggesting the involvement of inflammation. Furthermore, patients with a high adiponectin level (> 6.2 μg/ml) had lower Hb, LVEF, grip strength, knee extension strength, anterior mid-thigh muscle thickness, and SMI, compared to those with a low adiponectin level (< 6.2 μg/ml, the cut-off value of the ROC curve). On the other hand, the patients with a high adiponectin level had greater BNP, LAD, E/e', and LAVI, compared to those with a low adiponectin level. The serum concentration of TNFα in patients with a high adiponectin level was significantly greater than in those with a low adiponectin level. The present study also showed that the circulating adiponectin level was inversely correlated with nutritional factors (Hb, albumin). In contrast, a positive association between serum leptin levels and albumin was observed, suggesting that elevated serum leptin levels, but not adiponectin, are associated with good nutritional status. This is compatible with previous studies showing that leptin is a biological marker for

evaluating malnutrition [53,54]. Thus, it is likely that adiponectin plays an essential role in impaired metabolic signaling that is linked to heart failure progression including inflammation, muscle wasting, and poor nutrition in CVD patients receiving cardiovascular surgery, in contrast with leptin.

Some limitations of our study need consideration. First, because it was a cross-sectional study, the results did not imply causality. Second, the study had a small number of CVD patients undergoing different types of cardiovascular surgery and there were no CVD patients without surgery. Therefore, our findings are not necessarily applicable to the general population of CVD patients. In addition, most of the subjects had medical treatment. The use of drugs such as β-blockers [55], ACE-I, and ARB might have affected serum adipokine levels and metabolic profiles. Therefore, further detailed analyses in a larger number of CVD patients are required to clarify the pathophysiological significance of leptin and adiponectin.

In conclusion, we have shown that leptin and adiponectin may play a role in cardiac remodeling in CVD patients receiving cardiovascular surgery. And, adiponectin appears to be a marker of impaired metabolic signaling that is linked to heart failure progression including inflammation, poor nutrition, and muscle wasting in CVD patients receiving cardiovascular surgery as illustrated in Fig 2. In addition, leptin and adiponectin may be a useful biomarker for the operative risk in CVD patients receiving cardiovascular surgery, but the further studies are needed to clarify this possibility.

## Supporting information

**S1 Fig. Correlations between serum leptin and adiponectin levels and the echocardiographic findings.** Relationships between the echocardiographic findings (LAD (A), LAVI (B), E/e'(C)) and serum leptin (Aa, Ba, Ca) and adiponectin levels (Ab, Bb, Cb) *$P<0.05$, **$P<0.01$, ***$P<0.001$.
(TIF)

## Acknowledgments

We would like to thank cardiovascular surgery doctors and staffs for taking blood samples. We would also like to thank Mr. Satoshi Katayanagi for assistance with statistical analysis, and for the review and approval of the data.

## Author Contributions

**Data curation:** Ikuko Shibasaki, Hironaga Ogawa, Yuusuke Takei, Shigeru Toyoda, Shichiro Abe.

**Formal analysis:** Tatsuya Sawaguchi, Toshiaki Nakajima.

**Funding acquisition:** Toshiaki Nakajima.

**Investigation:** Toshiaki Nakajima.

**Methodology:** Tatsuya Sawaguchi, Akiko Haruyama, Takaaki Hasegawa, Takafumi Nakajima, Hiroyuki Kaneda, Takuo Arikawa, Syotaro Obi, Masashi Sakuma.

**Project administration:** Toshiaki Nakajima.

**Supervision:** Fumitaka Nakamura, Hirotsugu Fukuda, Teruo Inoue.

**Writing – original draft:** Tatsuya Sawaguchi, Toshiaki Nakajima.

**Writing – review & editing:** Toshiaki Nakajima.

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
