## [Decision Letter · Decision Letter 0]

6 Sep 2019

PONE-D-19-20823

Association of serum leptin and adiponectin concentrations with echocardiographic parameters and pathophysiological states in patients with cardiovascular disease receiving cardiovascular surgery

PLOS ONE

Dear Dr Nakajima,

Thank you for submitting your manuscript to PLOS ONE. After careful consideration, we feel that it has merit but does not fully meet PLOS ONE’s publication criteria as it currently stands. Therefore, we invite you to submit a revised version of the manuscript that addresses the points raised during the review process.

Two experts raised concerns and you should focus on clinical implications and avoid redundancy in data presentation.

We would appreciate receiving your revised manuscript by Oct 21 2019 11:59PM. To enhance the reproducibility of your results, we recommend that if applicable you deposit your laboratory protocols in protocols.io, where a protocol can be assigned its own identifier (DOI) such that it can be cited independently in the future. For instructions see: http://journals.plos.org/plosone/s/submission-guidelines#loc-laboratory-protocols

We look forward to receiving your revised manuscript.

Kind regards,

Tatsuo Shimosawa, M.D., Ph.D.

Academic Editor

PLOS ONE

Journal Requirements:

2. "We noticed you have some minor occurrence of overlapping text with the following previous publications, which needs to be addressed:

https://journals.plos.org/plosone/article?id=10.1371%2Fjournal.pone.0201499

https://synapse.koreamed.org/DOIx.php?id=10.3349/ymj.2012.53.1.91

In your revision ensure you cite all your sources (including your own works), and quote or rephrase any duplicated text outside the methods section. Further consideration is dependent on these concerns being addressed.

3. A reworded financial disclosure for the online submission form which states specifically whether the funders played any role in the study.

4. Please state the participants age range (in addition to their mean age) and any inclusion and exclusion criteria used during participant recruitment.

5. Data availability issue. In your statement you say "All data are fully available without restriction", but as we explain in http://journals.plos.org/plosone/s/data-availability#loc-faqs-for-data-policy you should provide the individual data points behind means, medians and variance measures presented in the results, tables and figures, and not just those summary statistics. Please provide these underlying participant-level data in a supporting information file or public repository, taking care not to include identifying information (see http://www.bmj.com/content/340/bmj.c181.long); if these data cannot be publicly deposited or included in the supporting information, e.g. due to patient privacy, legal reasons, or being provided by a third party, please explain why and explain how researchers may access them. Note that authors should not be the sole named individuals responsible for ensuring data access.

6. Please provide additional details regarding participant consent. In the ethics statement in the Methods and online submission information, please ensure that you have specified (1) whether consent was informed and (2) what type you obtained (for instance, written or verbal, and if verbal, how it was documented and witnessed). If your study included minors, state whether you obtained consent from parents or guardians. If the need for consent was waived by the ethics committee, please include this information.

7. We note that you have included the phrase “data not shown” in your manuscript. Unfortunately, this does not meet our data sharing requirements. PLOS does not permit references to inaccessible data. We require that authors provide all relevant data within the paper, Supporting Information files, or in an acceptable, public repository. Please add a citation to support this phrase or upload the data that corresponds with these findings to a stable repository (such as Figshare or Dryad) and provide and URLs, DOIs, or accession numbers that may be used to access these data. Or, if the data are not a core part of the research being presented in your study, we ask that you remove the phrase that refers to these data.

Reviewers' comments:

Reviewer's Responses to Questions

**Comments to the Author**

1. Is the manuscript technically sound, and do the data support the conclusions?

Reviewer #1: Partly

Reviewer #2: No

2. Has the statistical analysis been performed appropriately and rigorously? 

Reviewer #1: Yes

Reviewer #2: Yes

3. Have the authors made all data underlying the findings in their manuscript fully available?

Reviewer #1: Yes

Reviewer #2: Yes

4. Is the manuscript presented in an intelligible fashion and written in standard English?

Reviewer #1: Yes

Reviewer #2: Yes

5. Review Comments to the Author

Reviewer #1: In this manuscript, the authors examined preoperative serum leptin and adiponectin levels, and their relationships to pathophysiological states, parameters of blood tests, and echocardiographic parameters in patients undergoing cardiac surgery. The authors demonstrated that multiple parameters and biomarkers that correlated with leptin and adiponectin concentrations. They also found that serum adiponectin concentrations were associated with sarcopenia, with a putative cut-off value of 4.94 mcg/ml. They concluded that leptin may be a cardioprotective adipokine that reduces cardiac remodeling without muscle wasting, whereas adiponectin plays in cardiac dysfunction and impaired metabolic signaling that us linked to heart failure progression.

First of all, I would like to congratulate the authors on their interesting paper, in which they extensively analyzed preoperative leptin and adiponectin. However, it raises several important concerns that need to be addressed:

1. Cardiac functions and echocardiographic findings are strongly influenced by conditions and the severity of cardiac diseases. As the authors stated in limitation, the patients’ underlying backgrounds, general and cardiac conditions and medications vary, and their adiponectin and leptin levels were determined only preoperatively. Therefore, it is unclear whether leptin and adiponectin play causative roles in CVD. In other words, the authors should be careful of leading conclusions that these parameters are “cardioprotective” or “cardiodepressive” in their patient series. For example, BNP is the most reliable biomarker for cardiac strain and the severity of heart failure, and it is known to work as a potential cardioprotective peptide. The authors should carefully interpret the data, and explain why they concluded leptin might be “cardioprotective” and reduces cardiac remodeling, as opposed to previous reports as well as their background. Readers would be more interested in whether leptin and adiponectin work as potential biomarkers of the severity of cardiac diseases, and how they change after surgery.

2. They also presented the relationship between the adipokines and patients’ physical status and sarcopenia. This is a different story from cardiac functions but I personally find this part rather interesting. Since there are many questions raised in this context, I would suggest the authors present the cardiac part and sarcopenia in different papers. Otherwise, the authors may present a diagram showing putative interactions between cardiac diseases, cardioprotections and sarcopenia in Discussion. Since age is a cofounder of both adiponectin (in their data) and sarcopenia, is it possible a positive correlation of adiponectin with sarcopenia is due to age? Were there any relationship between gender and the prevalence of sarcopenia? The authors should clarify whether the cut-off value is applied regardless of age?　Is it applied to both male and female?

3. Some data they presented were redundant, which may have made the paper much more complicating. For example, both Figure 1 and 2 were duplicates of Table 3. I am not sure if it is necessary to present Model1-3 in Table 4. The authors presented relationships between serum adiponectin levels and GDF-15, TNF-alpha, d-ROMs. I would not find the data necessary, unless the authors examine potential roles of the relatively unfamiliar markers in their patient series.

Minor point: TNPα(p11, line 15) is misspelled.

Reviewer #2: In this study, the authors aimed to assess the relationship between serum leptin and adiponectin levels and echocardiographic parameters and pathophysiological states in patients with cardiovascular disease (CVD) receiving cardiovascular surgery.

1. Cardiac dysfunction was multifactorial and caused by various kind of cardiac disease, including heart valvular disease, coronary artery disease, hypertension, diabetes, and others. I find it difficult to see the clinical implication in a comparison between serum leptin, adiponectin levels, and echocardiographic parameters.

2. They measured inflammatory markers. However, they might be influenced by not only metabolic inflammation but by open-heart surgery, per se.

3. Too much data make difficult to understand their hypothesis. I encourage the authors to focus on data they need to prove their hypothesis.

6. PLOS authors have the option to publish the peer review history of their article (what does this mean?). If published, this will include your full peer review and any attached files.

Reviewer #1: No

Reviewer #2: No

---

## [Author Response · Author response to Decision Letter 0]

10 Oct 2019

Reply to Reviewer #1 

We greatly appreciate your careful attention to our manuscript and especially your excellent suggestions for improving the clarity and correctness of the message. We have corrected the paper as per your suggestions, and consider the revised manuscript much improved. 

Reviewer #1: In this manuscript, the authors examined preoperative serum leptin and adiponectin levels, and their relationships to pathophysiological states, parameters of blood tests, and echocardiographic parameters in patients undergoing cardiac surgery. The authors demonstrated that multiple parameters and biomarkers that correlated with leptin and adiponectin concentrations. They also found that serum adiponectin concentrations were associated with sarcopenia, with a putative cut-off value of 4.94 mcg/ml. They concluded that leptin may be a cardioprotective adipokine that reduces cardiac remodeling without muscle wasting, whereas adiponectin plays in cardiac dysfunction and impaired metabolic signaling that us linked to heart failure progression.

First of all, I would like to congratulate the authors on their interesting paper, in which they extensively analyzed preoperative leptin and adiponectin. However, it raises several important concerns that need to be addressed:

1. Cardiac functions and echocardiographic findings are strongly influenced by conditions and the severity of cardiac diseases. As the authors stated in limitation, the patients’ underlying backgrounds, general and cardiac conditions and medications vary, and their adiponectin and leptin levels were determined only preoperatively. Therefore, it is unclear whether leptin and adiponectin play causative roles in CVD. In other words, the authors should be careful of leading conclusions that these parameters are “cardioprotective” or “cardiodepressive” in their patient series. For example, BNP is the most reliable biomarker for cardiac strain and the severity of heart failure, and it is known to work as a potential cardioprotective peptide. The authors should carefully interpret the data, and explain why they concluded leptin might be “cardioprotective” and reduces cardiac remodeling, as opposed to previous reports as well as their background. 

#1) Answer. Thank you very much for your suggestions. We absolutely agree with your opinions. As described in discussion, the results did not imply causality, because it was a cross-sectional study. Therefore, we are not able to conclude that these adipokines are cardioprotective or cardiodepressive peptide. We changed the conclusion, and mentioned about it in discussions. 

Line 59, line 445, line 548 “These results suggest that leptin and adiponectin may play a role in cardiac remodeling in CVD patients receiving cardiovascular surgery.” 

Line 474. Thus, the present study provides the first evidence showing that leptin may be a cardioprotective adipokine that reduces cardiac remodeling in non-obese CVD patients receiving cardiovascular surgery. Thus, lower leptin levels may be associated with the loss of the protective effects of this adipokine, as reported in patients with heart failure [39]. However, the further studies using a large number of patients are required to clarify this possibility in cardiovascular surgery patients. 

Readers would be more interested in whether leptin and adiponectin work as potential biomarkers of the severity of cardiac diseases, and how they change after surgery.

#) Answer: We absolutely agree with you. However, we did not evaluate the changes of these biomarkers after surgery. We only mentioned about it in discussions.

Line 552. In addition, leptin and adiponectin may be a useful biomarker for the operative risk in CVD patients receiving cardiovascular surgery, but the further studies are needed to clarify this possibility.

2. They also presented the relationship between the adipokines and patients’ physical status and sarcopenia. This is a different story from cardiac functions but I personally find this part rather interesting. Since there are many questions raised in this context, I would suggest the authors present the cardiac part and sarcopenia in different papers. Otherwise, the authors may present a diagram showing putative interactions between cardiac diseases, cardioprotections and sarcopenia in Discussion.　

#) Answer. Thank you very much for your suggestion. We presented a diagram in Discussion. 

Figure 2. Leptin and adiponectin in CVD receiving cardiovascular surgery. Illustration summarizing the major findings described in the present study. Adiponectin has metabolic benefits, while leptin has metabolic risks. On the other hand, both adiponectin and leptin play a role in cardiac remodeling. And, adiponectin appears to be a marker of impaired metabolic signaling that is linked to heart failure progression including malnutrition, inflammation, and muscle wasting (cachexia, sarcopenia), especially in males, in contrast to leptin associating with well-nutrition and elevated muscle mass. 

Cardiac cachexia and muscle wasting (sarcopenia) have been reported to lead to increased levels

Since age is a cofounder of both adiponectin (in their data) and sarcopenia, is it possible a positive correlation of adiponectin with sarcopenia is due to age? Were there any relationship between gender and the prevalence of sarcopenia? The authors should clarify whether the cut-off value is applied regardless of age? Is it applied to both male and female?

Answer: Thank you very much for your suggestion. We have re-analyzed the data about the relationships between sarcopenia and adiponectin/leptin concentration in both males and females. As your suggestion, there were major differences between males and females. Therefore, we changed the data about it. 

Table 6 shows the relationships between serum adipokine (leptin and adiponectin) levels, physical function, and the BIA findings in both males and females. The serum leptin level was strongly positively correlated with body fat volume and body fat percentage in both males and females. In males, it had a positive correlation with skeletal muscle volume and SMI. On the other hand, adiponectin had a negative correlation with anterior mid-thigh muscle thickness, skeletal muscle volume and SMI in men, and it tended to correlate with body fat volume. In females, it had a weak negative correlation with SMI, and it tended negatively to correlate with skeletal muscle volume, and body fat volume. 

Table 6. Relationships between serum adipokine (leptin and adiponectin) levels, physical function, and the BIA findings in both males and females

Logistic regression analysis is used to obtain odds ratio in the presence of the explanatory variable (serum leptin and adiponectin levels) for sarcopenia in both males and females. Serum adiponectin level was an independent predictive factor in males (p = 0.037; odds ratio, 1.119; 95% confidence interval [CI] 1.007-1.124) for sarcopenia even after adjusted by age. However, serum leptin level was not an independent predictor factor (p = 0.440). In females, neither adiponectin nor leptin level was an independent predictive factor (p = 0.955 for adiponectin: p = 0.405 for leptin) for sarcopenia, adjusted by age. An ROC curve was plotted to identify the optimal cut-off level of the serum adiponectin concentration to detect sarcopenia in males, as shown in Fig. 1. To generate the ROC curve, different adiponectin cut-off values were used to predict sarcopenia with true positives on the vertical axis (sensitivity) and false-positives (1-specificity) on the horizontal axis. The area under the curve (AUC) was 74.7%. Sensitivity and specificity were 84.2%, and 64%, respectively. The optimal cut-off value was 6.2 μg/ml. 

Figure 1. ROC curve to identify the optimal cut-off level of the serum concentration of adiponectin to detect sarcopenia in males

To generate the ROC curve shown, different adiponectin cut-off values were used to predict sarcopenia with true positives on the vertical axis (sensitivity) and false positives (1-specificity) on the horizontal axis. 

Table 7 summarizes the comparative data from patients with low adiponectin levels (< 6.2 μg/ml, the cut-off value from the above ROC curve) and those with high adiponectin levels (> 6.2 μg/ml). As shown in Table 7, patients with high adiponectin levels had lower eGFR, Hb, TG, LVEF, grip strength, knee extension strength, and SMI, compared to those with low adiponectin levels. On the other hand, they had higher BNP, LAD, E/e’, and LAVI. The serum concentrations of TNFα in patients with high adiponectin levels were significantly greater than those in patients with low adiponectin levels (TNFα: 1.50 ± 0.60 pg/ml vs. 1.00 ± 0.42 pg/ml, p < 0.01). There were no significant differences of serum leptin level in both groups.

3. Some data they presented were redundant, which may have made the paper much more complicating. For example, both Figure 1 and 2 were duplicates of Table 3. I am not sure if it is necessary to present Model1-3 in Table 4. The authors presented relationships between serum adiponectin levels and GDF-15, TNF-alpha, d-ROMs. I would not find the data necessary, unless the authors examine potential roles of the relatively unfamiliar markers in their patient series.

Answer: Thank you very much for your comments.

Figure 1 and 3 were depleted. Figure 2 only was removed to the supplement Figure 1.

Model 1-3 in table 4 was also depleted in the revised version of our manuscript.

The data of GDF-15 and d-ROMs were deleted. 

Minor point: TNPα　(p11, line 15) is misspelled.

Answer. I corrected it.

Reply to Reviewer #2 

We greatly appreciate your careful attention to our manuscript and especially your excellent suggestions for improving the clarity and correctness of the message. We have corrected the paper as per your suggestions, and consider the revised manuscript much improved. 

In this study, the authors aimed to assess the relationship between serum leptin and adiponectin levels and echocardiographic parameters and pathophysiological states in patients with cardiovascular disease (CVD) receiving cardiovascular surgery.

1. Cardiac dysfunction was multifactorial and caused by various kind of cardiac disease, including heart valvular disease, coronary artery disease, hypertension, diabetes, and others. I find it difficult to see the clinical implication in a comparison between serum leptin, adiponectin levels, and echocardiographic parameters.

Answer: Thank you very much for your comments. As shown in limitation of the Discussion, the study had CVD patients undergoing different types of cardiovascular surgery. We have shown the additional detailed data of adiponectin and leptin on echocardiographic data. And, we added the following sentences into discussion.

Line 552. In addition, leptin and adiponectin may be a useful biomarker for the operative risk in CVD patients receiving cardiovascular surgery, but the further studies are needed to clarify this possibility.

Table 5. Multiple linear regression analysis of the serum leptin and adiponectin concentration and the echocardiographic findings

2. They measured inflammatory markers. However, they might be influenced by not only metabolic inflammation but by open-heart surgery, per se.

Answer. This study investigated the preoperative data. 

3. Too much data makes difficult to understand their hypothesis. I encourage the authors to focus on data they need to prove their hypothesis.

Answer. Thank you very much for your comments. I have deleted the part of the data, and as the another referee’s suggestion, we presented a diagram in Discussion. 

Figure 2. Leptin and adiponectin in CVD receiving cardiovascular surgery. Illustration summarizing the major findings described in the present study. Adiponectin has metabolic benefits, while leptin has metabolic risks. On the other hand, both adiponectin and leptin play a role in cardiac remodeling. And, adiponectin appears to be a marker of impaired metabolic signaling that is linked to heart failure progression including malnutrition, inflammation, and muscle wasting (cachexia, sarcopenia), especially in males, in contrast to leptin associating with well-nutrition and elevated muscle mass.

---

## [Decision Letter · Decision Letter 1]

24 Oct 2019

PONE-D-19-20823R1

Association of serum leptin and adiponectin concentrations with echocardiographic parameters and pathophysiological states in patients with cardiovascular disease receiving cardiovascular surgery

PLOS ONE

Dear Dr Nakajima,

Thank you for submitting your manuscript to PLOS ONE. After careful consideration, we feel that it has merit but does not fully meet PLOS ONE’s publication criteria as it currently stands. Therefore, we invite you to submit a revised version of the manuscript that addresses the points raised during the review process.

Please consider description on statistical analysis in method section. Also figure legend should be modified to make it easier to follow.

We would appreciate receiving your revised manuscript by Dec 08 2019 11:59PM. To enhance the reproducibility of your results, we recommend that if applicable you deposit your laboratory protocols in protocols.io, where a protocol can be assigned its own identifier (DOI) such that it can be cited independently in the future. For instructions see: http://journals.plos.org/plosone/s/submission-guidelines#loc-laboratory-protocols

We look forward to receiving your revised manuscript.

Kind regards,

Tatsuo Shimosawa, M.D., Ph.D.

Academic Editor

PLOS ONE

Reviewers' comments:

Reviewer's Responses to Questions

**Comments to the Author**

1. If the authors have adequately addressed your comments raised in a previous round of review and you feel that this manuscript is now acceptable for publication, you may indicate that here to bypass the “Comments to the Author” section, enter your conflict of interest statement in the “Confidential to Editor” section, and submit your "Accept" recommendation.

Reviewer #1: All comments have been addressed

Reviewer #2: All comments have been addressed

2. Is the manuscript technically sound, and do the data support the conclusions?

Reviewer #1: Yes

Reviewer #2: Yes

3. Has the statistical analysis been performed appropriately and rigorously? 

Reviewer #1: I Don't Know

Reviewer #2: Yes

4. Have the authors made all data underlying the findings in their manuscript fully available?

Reviewer #1: Yes

Reviewer #2: Yes

5. Is the manuscript presented in an intelligible fashion and written in standard English?

Reviewer #1: (No Response)

Reviewer #2: Yes

6. Review Comments to the Author

Reviewer #1: I thank the authors for revising the manuscript. Most of the comments and corrections that they made are appropriate. Before accepting the paper, I suggest a few more points as below:

1) I recommend that a statistician reviews and approves the data, and the authors describe so in M & M.

2) The authors showed potentially different roles of Leptin and Adiponectin in “Cardiac remodeling”. I am not sure if they intended to mean so in Fig.2. I suggest them to describe more clearly on this point (ex. Adiponectin may worsen atrial strain and diastolic dysfunction). The Figure legend should be what is like “A Putative role of Leptin and Adiponectin in…”.

Reviewer #2: (No Response)

7. PLOS authors have the option to publish the peer review history of their article (what does this mean?). If published, this will include your full peer review and any attached files.

Reviewer #1: No

Reviewer #2: No

---

## [Author Response · Author response to Decision Letter 1]

24 Oct 2019

Reply to Reviewer #1 

We greatly appreciate your careful attention to our manuscript and especially your excellent suggestions for improving the clarity and correctness of the message. We have corrected the paper as per your suggestions, and consider the revised manuscript much improved. 

Reviewer #1: I thank the authors for revising the manuscript. Most of the comments and corrections that they made are appropriate. Before accepting the paper, I suggest a few more points as below:

1) I recommend that a statistician reviews and approves the data, and the authors describe so in M & M.

Answer: Thank you very much for your suggestion. I described it in “Acknowledgments“。

Acknowledgments 

We would also like to thank Mr. Satoshi Katayanagi for assistance with statistical analysis, and for the review and approval of the data. 

2) The authors showed potentially different roles of Leptin and Adiponectin in “Cardiac remodeling”. I am not sure if they intended to mean so in Fig.2. I suggest them to describe more clearly on this point (ex. Adiponectin may worsen atrial strain and diastolic dysfunction). The Figure legend should be what is like “A Putative role of Leptin and Adiponectin in…”

Answer. Thank you very much for your comments. I corrected Figure 2 legend as follows.

Figure 2.A putative role of leptin and adiponectin in CVD patients receiving cardiovascular surgery 

Illustration summarizing the major findings described in the present study. Adiponectin has metabolic benefits, while leptin has metabolic risks. On the other hand, both adiponectin and leptin play a role in cardiac remodeling. Leptin appears to be associated with lower LV mass, and LA size. In contrast, adiponectin may involve diastolic dysfunction, and appears to be a marker of impaired metabolic signaling that is linked to heart failure progression including malnutrition, inflammation, and muscle wasting (cachexia, sarcopenia), especially in males, in contrast to leptin associating with well-nutrition and elevated muscle mass.

---

## [Decision Letter · Decision Letter 2]

28 Oct 2019

Association of serum leptin and adiponectin concentrations with echocardiographic parameters and pathophysiological states in patients with cardiovascular disease receiving cardiovascular surgery

PONE-D-19-20823R2

Dear Dr. Nakajima,

We are pleased to inform you that your manuscript has been judged scientifically suitable for publication and will be formally accepted for publication once it complies with all outstanding technical requirements.

With kind regards,

Tatsuo Shimosawa, M.D., Ph.D.

Academic Editor

PLOS ONE

Additional Editor Comments (optional):

Reviewers' comments:

Reviewer's Responses to Questions

**Comments to the Author**

1. If the authors have adequately addressed your comments raised in a previous round of review and you feel that this manuscript is now acceptable for publication, you may indicate that here to bypass the “Comments to the Author” section, enter your conflict of interest statement in the “Confidential to Editor” section, and submit your "Accept" recommendation.

Reviewer #1: All comments have been addressed

2. Is the manuscript technically sound, and do the data support the conclusions?

Reviewer #1: Yes

3. Has the statistical analysis been performed appropriately and rigorously? 

Reviewer #1: Yes

4. Have the authors made all data underlying the findings in their manuscript fully available?

Reviewer #1: Yes

5. Is the manuscript presented in an intelligible fashion and written in standard English?

Reviewer #1: Yes

6. Review Comments to the Author

Reviewer #1: (No Response)

7. PLOS authors have the option to publish the peer review history of their article (what does this mean?). If published, this will include your full peer review and any attached files.

Reviewer #1: No

---

## [Editor Report · Acceptance letter]

1 Nov 2019

PONE-D-19-20823R2 

Association of serum leptin and adiponectin concentrations with echocardiographic parameters and pathophysiological states in patients with cardiovascular disease receiving cardiovascular surgery 

Dear Dr. Nakajima:

I am pleased to inform you that your manuscript has been deemed suitable for publication in PLOS ONE. Congratulations! Your manuscript is now with our production department. 

With kind regards,

on behalf of

Prof. Tatsuo Shimosawa 

Academic Editor

PLOS ONE